# Self-Reproduction and Darwinian Evolution in Autocatalytic Chemical Reaction Systems

**DOI:** 10.3390/life11040308

**Published:** 2021-04-01

**Authors:** Sandeep Ameta, Yoshiya J. Matsubara, Nayan Chakraborty, Sandeep Krishna, Shashi Thutupalli

**Affiliations:** 1Simons Centre for the Study of Living Machines, National Centre for Biological Sciences, Tata Institute of Fundamental Research, Bangalore 560065, India; 2International Centre for Theoretical Sciences, Tata Institute of Fundamental Research, Bangalore 560089, India

**Keywords:** origins of life, autocatalytic set, emergence, Darwinian evolution, self-reproduction

## Abstract

Understanding the emergence of life from (primitive) abiotic components has arguably been one of the deepest and yet one of the most elusive scientific questions. Notwithstanding the lack of a clear definition for a living system, it is widely argued that heredity (involving self-reproduction) along with compartmentalization and metabolism are key features that contrast living systems from their non-living counterparts. A minimal living system may be viewed as “a self-sustaining chemical system capable of Darwinian evolution”. It has been proposed that autocatalytic sets of chemical reactions (ACSs) could serve as a mechanism to establish chemical compositional identity, heritable self-reproduction, and evolution in a minimal chemical system. Following years of theoretical work, autocatalytic chemical systems have been constructed experimentally using a wide variety of substrates, and most studies, thus far, have focused on the demonstration of chemical self-reproduction under specific conditions. While several recent experimental studies have raised the possibility of carrying out some aspects of experimental evolution using autocatalytic reaction networks, there remain many open challenges. In this review, we start by evaluating theoretical studies of ACSs specifically with a view to establish the conditions required for such chemical systems to exhibit self-reproduction and Darwinian evolution. Then, we follow with an extensive overview of experimental ACS systems and use the theoretically established conditions to critically evaluate these empirical systems for their potential to exhibit Darwinian evolution. We identify various technical and conceptual challenges limiting experimental progress and, finally, conclude with some remarks about open questions.

## 1. Introduction: Autocatalysis and the Emergence of Life

A cell, the widely accepted “unit of life”, is able to propagate by transferring templated genetic information to daughter cells, aided and buffered by a plethora of complex processes and catalytic machinery. In contrast, “minimal life” may be viewed as a self-sustaining chemical system capable of undergoing Darwinian evolution, i.e., for heredity, variation, and selection to play out [1]. This immediately raises the following questions: (i) What constitutes such a self-sustaining chemical system? (ii) What properties of such a system allow for heredity, i.e., the stable propagation of information (via a self-reproducing unit)? (iii) What features of the chemical system and the heredity mechanism allow for chemical dynamics resembling Darwinian evolution? Information in a self-sustained chemical system can be propagated either as a code, templated in a complex polymer, or as the composition of a set of chemical reactions [2]. While self-reproduction and the potential for Darwinian evolution are fairly well-established for a template-based replication system, how such a templated system, with sufficient reproductive fidelity and complex function, could have emerged from a primitive “messy” chemistry (prebiotic chemical soup) remains an open question [3]. In contrast to a templated replication system, a self-reproducing unit might also be identified by a unique collective composition of a set of chemical reactions (Note: we use the term replication only in the context of a monomer-by-monomer copy of the sequence information occurring via template-directed polymerization and the term self-reproduction otherwise.). In this context, autocatalytic sets of chemical reactions (ACSs), comprising various chemical species, with cooperative catalytic interactions have been proposed [2,4,5,6,7,8] as an intermediate stage (Figure 1) of chemical evolution (Note: while our focus here is on ACSs, the demonstration of intermediary chemical and evolutionary dynamics has also been explored in chemical systems that lack autocatalysis [9]).

Autocatalytic systems have been the subject of intense study over the years in elucidating the steps towards the origins of life [10,11,12,13,14]. Some of the questions about ACSs that continue to be addressed, both theoretically and experimentally, include the following: (i) How can an ACS emerge from a prebiotic “messy chemical” soup? (ii) Under what conditions can stable chemical identity of the ACS be established? (iii) Can such an identity (unit) reproduce heritably? (iv) How does (heritable) variation of the chemical identities arise? (v) Can such a variation lead to differential fitness and result in Darwinian evolution? (vi) Can the chemical dynamics of an ACS give rise to templated replication? (Note: metabolism-first scenarios hypothesize that the hereditary mechanisms of ACSs could later have been “taken over” by higher fidelity replication systems [15,16,17].). While theoretical progress has been made on some of these questions, experiments are either still catching up or altogether lacking. In this review, we critically examine that available experimental systems which form autocatalytic reaction networks, specifically for their potential for evolution. We begin with an overview of the theoretical ideas that laid some of the foundations for the conceptualization and analysis of autocatalytic reaction networks and their dynamics.

**Figure 1 life-11-00308-f001:**
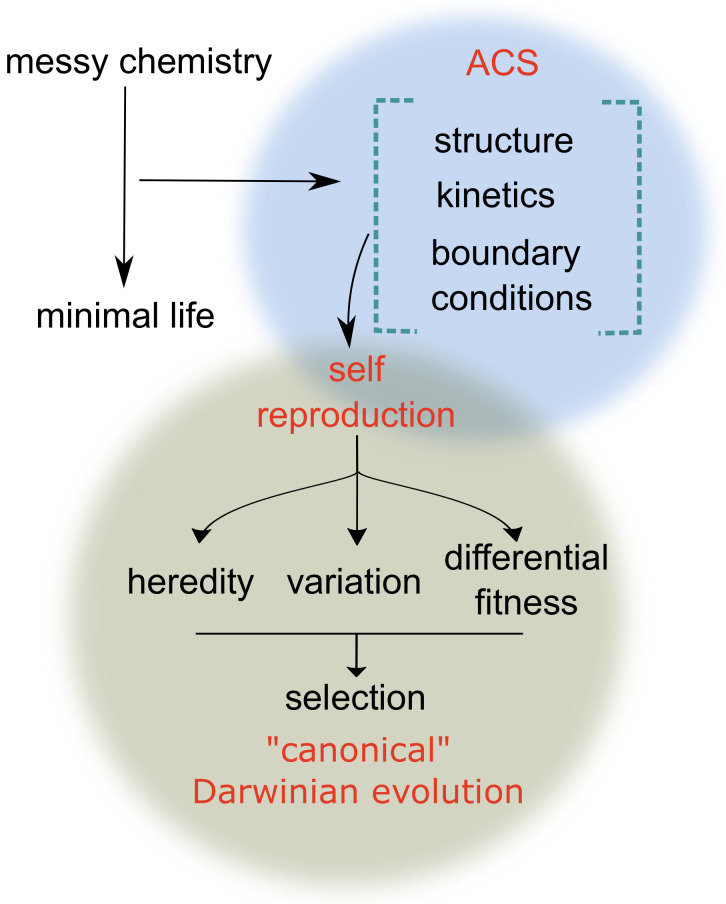
Schematic overview of autocatalytic sets of chemical reactions (ACSs) as a potential intermediate stage between prebiotic “messy chemistry” and the emergence of minimal life. In this overview, we define ACSs based on their reaction network structure and reaction kinetics and evaluate the (boundary and other) conditions for their self-reproduction and canonical Darwinian evolution, as defined by the specific characteristics of variation, heredity, and differential fitness [18,19].

## 2. Theoretical Perspectives on the Emergence and Evolvability of Autocatalytic Systems

Typically, an autocatalytic system is imagined as a chemical reaction network and its population dynamics are modeled using deterministic or stochastic differential equations. We first define an autocatalytic system in terms of reaction network architectures and kinetics (Section 2.1) and then discuss the constraints necessary for self-reproduction of an ACS (Section 2.2). Subsequently, we review studies on the spontaneous emergence of self-reproducing systems (Section 2.3) and evaluate the requirements for an autocatalytic system to undergo canonical Darwinian evolution, specifically the aspects of variation, heredity, and differential fitness (Section 2.4).

### 2.1. Defining Autocatalytic Chemical Reaction Networks

**Static network architectures of autocatalytic sets of reactions**: Autocatalytic reaction networks, originally proposed by Kauffman [7,20], in a broad sense, comprise of nodes—molecules—and directed edges—catalyzed reactions that produce molecules from building blocks (“food”). In such a network, representing multiple reactions, an autocatalytic network is defined as a subnetwork where the catalysts required to produce each molecule (node) are present within the same subnetwork [8,21] (Note: at times, substrates that are ubiquitously available, or buffered, such as food molecules (introduced by Kauffman [7]), discussed shortly, may be omitted from this description). Such an ACS must contain a closed cycle of edges (e.g., Figure 2a), where all the members in the cycle are produced by reactions catalyzed by another molecule in the same cycle [8,22]. Such a closed cycle also appears in the so-called “hypercycle” model originally proposed by Eigen [11]. The cycle is comprised of multiple template molecules (or enzymes catalyzed by templates) catalyzing the synthesis of another molecule, altogether resulting in the replication of long template molecules. (Note: strictly, in the original context, a hypercycle is not introduced as an autocatalytic set since it is concerned with the replication of a templated molecule [23]. Further, as discussed later, in contrast with the Jain–Krishna model [8] (see Section 2.2), the reproduction of each molecule is a second-order autocatalytic reaction because it is associated with both the template and other catalytic molecules [23]. Although template-directed replication, as in their original context, is out of the scope of the present review, similar dynamics due to higher-order reaction could emerge in other autocatalytic systems even without template replication described later.)

To proceed further, especially with the view of developing mathematical frameworks, consider reaction networks in which the substrates of all the reactions are explicitly accounted for. Following Kauffman’s initial work, using computational simulations, a formal definition of such an autocatalytic network was given as follows [24]: an ACS is comprised of a food (substrate) set, a reaction set, and a molecule set; (i) each reaction is catalyzed by at least one molecule, and (ii) every molecule is produced by a series of reactions from food molecules. (e.g., Figure 2b). A more general definition of autocatalysis in a network uses reaction network stoichiometry [25,26,27,28,29]; a cycle of reactions (which are not necessarily catalytic) is defined as autocatalytic if the chemicals within the cycle increase stoichiometrically over time (e.g., Figure 2c). Such autocatalytic network structures have recently been identified in an empirical dataset representing (ancient) metabolic networks [30] and prebiotic chemistry networks [31].

The likelihood of the existence of an ACS in a randomly generated network has been studied under various assumptions [32,33,34,35,36]. These studies find that the likelihood of existence is typically an exponentially decreasing function of the size of an ACS. In contrast, for a reaction network of fixed size with random links, as the number of links (edges) increases, the probability of the existence of an ACS shows a phase transition—from 0 to 1—as the number of edges crosses a threshold. This is the same as the “percolation” transition [37], wherein a random graph goes from being composed of largely disconnected clusters of nodes to one giant connected cluster [38]. For example, consider a system with *M* species of molecules {X_1_, X_2_, …, X_*M*_} and reactions catalyzed by one of them such that
$$X_{i}+X_{j}⟶X_{k}+X_{j},$$
where a type X*_j_* catalyzes conversions of type X*_i_* into X*_k_* (e.g., Figure 2d). Here, X*_i_* and X*_j_* react with random probability ρ, and X*_k_* is chosen as the product from all types except X*_i_* and X*_j_*. This randomly directed network with reaction link density ρ percolates at a threshold $\rho^{*}∼1/M$ [39,40]: below this threshold, the formation of an ACS is not plausible; on the other hand, above the threshold, with high probability, an ACS comprising (almost) the entire set of molecular species is formed.

These structural definitions of an autocatalytic network neither take reaction kinetics into account nor include any information about the boundary conditions on the system. Next, we discuss the extension of these network models to kinetic ones and examine the chemical composition dynamics, specifically to explore the possibility of self-reproduction in ACSs.

**Reaction kinetics of ACSs**: We start by noting that a single autocatalytic reaction is one in which the product catalyzes its own synthesis (see, e.g., [41,42]) and the simplest autocatalytic system that converts a buffered or highly abundant substrate S into a catalyst X is of the type: $$S+X\overset{k_{a}}{⟶}2X.$$

Note, on the other hand, that a reaction of the type
$$S\overset{k_{b}}{⟶}X,$$
is not autocatalytic since it occurs without catalysis via the product of the reaction and is considered as a “background reaction”. That is, whenever the above catalyzed reaction occurs, the uncatalyzed one may also occur spontaneously. There are multiple interpretations of such a background reaction: (i) the activation energy is low enough for the reaction to proceed even without a catalyst, or (ii) other weak catalysts may assist the reaction to proceed, e.g., non-covalent complexes formed by the substrates [43]. Representing the amounts of S and X as *s* and *x*, respectively, the rate equation for the amount of X is then
$$\frac{dx}{dt}=s\left(k_{a}x+k_{b}\right),$$
where $k_{a}$ and $k_{b}$ are rate constants of the catalyzed and background reactions, respectively, and *s* is assumed to be sufficiently large and virtually constant. The first term on the right hand side denotes that part of the reaction where X is produced at a rate that accelerates with time. If the background reaction is negligible, X grows exponentially since its production rate is proportional to its own concentration. If, however, the background is dominant, X grows linearly as the production rate is constant. Therefore, a sigmoid-like curve of the dynamics of *x* is indicative of the presence of autocatalysis (Note: another way to confirm autocatalysis experimentally is to measure the kinetic parameters of the reactions by fixing the substrate concentrations and seeding by varying the product catalysts and by measuring the initial production rate of the product [44]).

Here, a few comments regarding the production rate are in order: the above scheme can be rewritten as $S\overset{r(x)}{⟶}X,$ i.e., the production rate r(x) depends on the amount *x* itself. If, for example, Michaelis–Menten-like kinetics are assumed for the catalytic reaction instead of mass action kinetics, the dependence of the production r(x) on *x* is sublinear—i.e., $r\left(x\right)∼x^{\alpha }$, where α is the order of the autocatalytic reaction and α<1—and the growth of *X* is sub-exponential. This behavior is seen in experiments using a template-directed reproduction scheme [44], where the dissociation of the template and the product is very slow and the reaction order α∼1/2 asymptotically [45]. While such sub-exponential reaction dynamics tend to appear due to mutual inhibition between reaction species, higher-order reaction terms could emerge due to cooperation among catalysts (or templates). If α>1, growth is hyperbolic (super-exponential)—a theoretical example of such super-exponential dynamics is that of a so-called “hypercycle” by Eigen [11,22]. Furthermore, higher-order autocatalysis has also been observed in systems using DNA [46] (Note: while higher-order growth and frequency-dependent selection of DNA sequences was shown in [46] using a theoretical model of autocatalytic templates based on ligation reactions, the experimental demonstration uses a highly evolved enzyme to enhance the reaction speeds and is not autocatalytic.) or peptide template-ligation systems [47]. Generally speaking, in reaction networks comprised of multiple catalyzed reaction steps, higher-order autocatalytic reaction dynamics could appear.

### 2.2. Self-Reproduction of Autocatalytic Systems

Broadly, the self-reproduction of an ACS can be viewed as the generation of a copy—from the substrates—of the autocatalytic chemical system due to the self-catalyzed reaction dynamics (and not due to background reactions). This generally depends on the “boundary conditions” on the system—sustained self-reproduction requires a continuous flux of chemicals and associated dissipation, i.e., out-of-equilibrium. Indeed, a defining characteristic of life is energy consumption [48], which arises naturally from the necessity for continuous self-reproduction. Such out-of-equilibrium situations can be realized due to the boundary conditions on the system, which may be of the following kinds: (i) continuous stirred-tank reactor (CSTR) or chemostat, (ii) multiple cycles of serial transfer, and (iii) generic compartmental dynamics such as cell growth and division. In a CSTR (chemostat), substrates are fluxed in at a constant rate and are kept well-stirred within the system where reactions occur. Various contents (substrates and products) flux out of the system at the same rate as the influx. Note that, in such a setup, the total mass of the reaction substrate is kept constant. In a serial-transfer protocol, reactions proceed for a certain fixed period, after which the system is diluted into a fresh batch and the cycle is then repeated. Another relevant boundary condition relates to that of a chemical system contained within a “compartment” (e.g., coacervates, lipid bilayer vesicles, localized regions on surfaces, and hydrodynamic flow structures) that are—in very broad terms—spatially localized chemistries that can grow and divide, similar to present cellular systems (Note: a CSTR may be viewed as a particular extreme case of a serial-transfer protocol or as a special case of compartment dynamics protocols, where the volume of the compartment grows strictly exponentially.). The self-reproduction of an ACS is evident, for example, if the chemical system is contained in a compartment—self-reproduction is then naturally associated with the exponential growth and division of the compartments. However, in other conditions, such as in a CSTR, what constitutes self-reproduction is not obvious at the outset.

Generally, in any of the above conditions, a chemical reaction system eventually reaches stationary stable states—attractors, in the language of dynamical systems (this need not be steady state with constant chemical concentrations—a dissipative chemical reaction system can also display various complex behaviors, such as oscillation, multi-stability, and chaotic dynamics and the notion of “stationary stable states” or attractors can be extended to these behaviors too [49,50,51]). In the present context, self-reproduction of autocatalytic networks is characterized as one such stationary state. In a CSTR condition, an autocatalytic network reaching a specific compositional steady state where its components are produced by reactions catalyzed by itself instead of background reactions is viewed as self-reproducing (Note: for example, in the bistable system in Section 2.3, we regard the system in the active state as self-reproducing but not in the inactive state. In the active state, the catalyzed reactions dominantly produce ACS components, while in the inactive state, the background reactions are dominant.).

Given an autocatalytic network, associated kinetics, and a suitable boundary condition, as discussed above, what determines whether the system can self-reproduce, i.e., make a copy of the chemical reaction system?

Consider the example of a kinetic model of an autocatalytic reaction network based on first-order reaction kinetics: $$S+X_{j}\overset{C_{ij}}{⟶}X_{i}+X_{j},$$
where X*_i_* is a molecular species, S is a substrate (the notation is similar to that used in the section above), and $c_{ij}$ is the rate of the reaction for X*_j_* catalyzing the production of X*_i_*. Examples of such systems could be theoretical models such as the Jain–Krishna model [8] or the GARD (Graded Autocatalysis Replication Domain) model [2,52], and experimental systems such as ones based on the *Azoarcus* ribozyme [53] or peptides [54]. Here, (for the boundary condition) assume that the system is chemostatted, i.e., a steady flux of substrates and chemicals is maintained (Note: in the experimental systems mentioned above, a steady flux is not maintained but rather an abundance of substrates is present at the start of the reactions.). In this situation, the kinetic equations for species $\left\{x_{1},x_{2},\ldots ,x_{M}\right\}$ are
$$\frac{dx_{i}}{dt}=s\sum_{j=1}^{M}c_{ij}x_{j}-dx_{i}.$$

These deterministic rate equations for $X_{i}$ have a single stable compositional solution (steady state), irrespective of the initial conditions, if and only if the reaction system has an autocatalytic cycle that is composed of reactions with nonzero $c_{ij}$ [8] (Note: strictly, this is only true if there is only one ACS in the system (which need not be just a cycle and could be a more complex ACS). However, if there are, for example, no ACSs or multiple disconnected ACSs, then one can have multiple states even with these simple linear equations.). Therefore, reaching a steady state comprising of a nontrivial, nonzero stable chemical composition represents the generation of a copy of the reaction network from substrates, i.e., self-reproduction of the ACS. If the differences among $c_{ij}$ are large enough—such as in a stochastic GARD model with a finite system size and randomly distributed kinetic parameters $c_{ij}$—the system cannot sustain self-reproduction because of the extinction of some molecular species. Extinction could result either due to large enough differences in the kinetic parameters or the stochasticity due to a small enough system size. On the other hand, in such a system, there could be multiple compositional states [2], with consequences for the evolvability of the network, which will be discussed in Section 2.4.

We now briefly remark on the conditions for self-reproduction in other situations, specifically involving higher-order reaction kinetics. In general, if the reaction kinetics are of order higher than unity, the compositional dynamics could show more complex behavior and instabilities.

First, we consider the example of a dynamical instability arising in a hypercycle network, viz., that of “parasites”. Let a hypercycle with two species have the following properties: one species catalyzes the reproduction of both types, but the second species lacks any catalytic ability, i.e., it is a “parasite”. If the reproduction rate of the parasite is faster than that of the catalyst, the parasite eventually dominates the system and reproduction of both species cannot sustain. Therefore, although the network has a closed cycle that potentially can reproduce all the species within it, this system cannot exhibit sustained self-reproduction without appropriate kinetic rates.

Next, we consider the example of the second-order kinetics of an autocatalytic system in a random catalytic network (e.g., a static network of the type shown in Figure 2d). Assume the existence of an ACS in the network, i.e., the reaction density ρ is above the percolation threshold, and a CSTR boundary condition. In this system, if the rates of substrate supply and dissipation are sufficiently high, the reaction kinetics biases the network composition towards species produced by the shortest reaction pathway from the food set (Figure 2d). Due to the nonlinearity in the kinetics, viz., second-order reaction kinetics, such a bias could result in the loss of autocatalysis altogether due to some of the catalytic species being too low in abundance, i.e., reactions catalyzed by species with small enough amounts do not occur at all. Therefore, there exists an upper limit on the dissipation/supply rate beyond which self-reproduction is no longer possible. A static autocatalytic network structure is not sufficient for self-reproduction; kinetic parameters such as the dissipation rate, reaction rate constants, etc. should also be appropriate.

Third, even if conditions (on network structure and kinetic parameters) for self-reproduction are met, there exists a certain threshold for the initial amount of catalysts, which separates the final compositions into catalytically active or inactive states [55]. The implication of these catalytically active and inactive states directly relates to the spontaneous emergence of an autocatalytic system, which we discuss next.

### 2.3. Spontaneous Emergence of Autocatalytic Systems

In general, the catalytic molecules comprising an autocatalytic system could themselves be quite complex molecules requiring other molecular complexes to catalyze their formation. Therefore, the seemingly “chicken or egg” dilemma regarding how autocatalytic systems could have first emerged remains a challenging open question in understanding the origins of life.

Oparin hypothesized [56] that the first pre-life system could have emerged as coacervates from a “chemical soup”—the coacervate compartment itself contains various organic molecules that could reproduce as a whole by mutual, unspecific interactions. Expanding on Oparin’s hypothesis, Dyson proposed a mathematical model discussing specifically the probability of the emergence of such a self-reproducing system [6,16]. In Dyson’s model, polymers (complex molecules) autocatalytically aid other polymers to be (catalytically) active. In an appropriate parameter regime of the model, the system exhibits a bistability between (i) a catalytically inactive state and (ii) a catalytically active state. The catalytically inactive state is one in which the overall fraction of catalytic polymers is low, and a catalytically active state is one in which this fraction is high. The transition from the inactive state to an active one is precisely related to the emergence of autocatalytic sets. Dyson compared the possibility of the emergence of the active catalytic state between two different scenarios for autocatalytic systems—(i) systems that reproduce themselves collectively but not accurately and (ii) systems that can replicate themselves with high fidelity—and concluded that the former type of system possibly emerged before the latter.

Jain and Krishna studied a similar model to that of Dyson by considering an evolving network of reactions (where species are added at random and are lost due to low abundance); they demonstrated the spontaneous emergence of an autocatalytic system that eventually spans the entire reaction network. However, the lack of chemical realism in both the models leads to the consideration of reactions with more “realistic” steps [7,57,58,59]. For example, Kauffman considered a model [7] of “binary polymers” (i.e., a “protein” comprised of 2 amino acid types). These polymer species can “cleave” and “ligate” (i.e., conjoin) each other through reactions catalyzed via other similar polymers to produce multiple other species. Similar to previously discussed models, consider that any such catalyzed reaction exists with a probability ρ. Kauffman showed that, starting from a food set of short polymers, longer and more diverse polymers are produced if ρ exceeds a certain threshold $\rho^{*}$ in the initial food sets—identical to the percolation threshold in a reaction network discussed in Section 2.1—because newly created polymers catalyze the production of the other new species, leading eventually to infinitely long polymers. However, it was found that the threshold probability $\rho^{*}$ far exceeds the observed values in any empirical system and that the food and shorter polymers should have high catalytic activity, relating again to the “chicken or egg” type of problem alluded to earlier. Bagley and Farmer also considered a model in which polymers ligate and cleave, and unlike Kauffman’s model, uncatalyzed background reactions are also taken into account [57]. Small amounts of novel long polymers are indeed produced via such background reactions; however, for a sustainable autocatalytic set containing such long polymers to emerge, the rate of the catalytic reaction should be far larger than the rate of background reactions. There have been a few proposals to remedy to this situation: (i) the so-called “nested ACS”, in which an ACS comprised of longer polymers, is reinforced by an ACS comprising of weak but short catalytic polymers [55]; (ii) stochastic fluctuations could lead to the emergence of ACS containing longer polymers [60]—here, the emergence of an ACS corresponds to a phase transition from the inactive to the active states in Dyson’s model, and the stochastic analysis further suggests an optimal size of the system (i.e., volume of the compartment containing the reaction system) for the possibility of such a transition; and (iii) template-assisted spontaneous ligation can lead to open-ended polymer elongation and autocatalysis [61,62,63,64].

Given these various scenarios for the spontaneous emergence of a complex autocatalytic reaction system, we next turn our attention to the possibilities of canonical Darwinian evolution of ACSs.

### 2.4. Darwinian Evolution of Autocatalytic Systems

The specific characteristics of Darwinian evolution that we discuss here are (i) variation, (ii) heredity, and (iii) differential fitness, following the formulation by Lewontin and Godfrey-Smith [18,19]. First, variation refers to a population of different types of reproducible units $X_{1},X_{2},\ldots$, where the $X_{i}$s are a coarse-grained representation of the reproducible unit, i.e., the units could either be the composition of a chemical reaction network, a template molecule(s), or other higher level entities such as protocells (compartments). Second, heredity refers to the type of reproducible unit inherited during reproduction i.e., $S+X_{i}\overset{\lambda_{i}}{⟶}2X_{i}$, where $\lambda_{i}$ is the reproduction rate (“fitness”) of the reproducible unit $X_{i}$. Third, if such units with variation and heredity, having differential fitness, share the same substrate (food) and are maintained out of equilibrium, “selection” of a few among the reproducible units could occur. Taken together, the three ingredients detailed here constitute “canonical” Darwinian evolution. For example, in an open flux CSTR condition, evolution could occur in the following way: rate equations for the amounts of $X_{i}$, i.e., $\chi_{i}$, are given by $\frac{d\chi_{i}}{dt}=s\lambda_{i}\chi_{i}-d\chi_{i},$ where *d* is the flux rate of the CSTR. Then, total mass conservation of substrates leads to resource competition among the reproducible units $X_{i}$, leading to the survival of the one with the fastest reproduction rate (i.e., the “fittest”) in the steady-state.

Before we proceed further, we provide a brief note on the nature of the reproducible units. As discussed earlier, unlike in a templated system, in which the information-carrying polymers themselves can be the units of reproduction, it is the chemical composition of autocatalytic networks (and not any one individual molecule) that is the unit of reproduction and hence of selection in any ensuing evolutionary dynamics. Variation arises in that the system could have different stable compositional states, which are inherited across generations, and such states could also be “mutable” i.e., de novo changes in the identities of reproductive units may occur. Such mutations could occur either stochastically or could arise as a combination of reaction kinetics, e.g., due to background reactions and boundary conditions. Here, it is important to distinguish heritable compositional variation from trivial compositional variation resulting as a consequence of boundary conditions, e.g., differences resulting simply due to the composition of food sets.

The principles of evolution and selection have been extensively studied for the case in which template molecules are the units of selection following Eigen’s formulation of the quasi-species model [11]. In contrast, there is no well-established theory for the evolution of autocatalytic networks [65]. Below, we extend our discussion of autocatalytic reaction systems considered so far and review what conditions on these systems allow for variation, heredity, and differential fitness.

**Catalytic network topology and the existence of units of selection**: First, we examine the conditions for the existence of units of selection [18,66]. A subnetwork that is autocatalytic in itself and not dependent on the rest of the whole network (which we refer to as a viable “core” network) could be a unit of selection. (Note: cores need not be completely separated from the rest of the network; e.g., outgoing links from a core do not affect its property as a unit of reproduction. Additionally, weak links between cores can be regarded as “background reactions” as we consider here.) The existence of several such subsets are essential for the emergence of heritable variation in chemical composition. As we have discussed earlier, some such subsets are formed in the ACS based on first-order reaction kinetics if there are large enough deviations in the kinetic parameters [2,67]. The importance of core networks in an ACS for multiple compositional states was pointed out also in the Jain–Krishna model [68], in a random hypercycle [69], and in Kauffman’s polymer model [14]. The condition that a random autocatalytic network has multiple separate core autocatalytic subnetworks depends on the assumptions of the models. For example, in a polymer autocatalytic network model [14], if the network connectivity is too small, there is no autocatalytic set, but if the connectivity is too large, the whole network is connected (via network percolation) and there are no isolated autocatalytic subnetworks (which can be units of the selection). We emphasize that the existence of a reproducible unit is not sufficient for the possibility of evolution but rather what is required is the existence of several types of such units. (Note: a percolated network is in some sense a single self-reproducing unit and therefore lacks the possibility for evolution. Note that, although a percolated network can have many subnetworks that are autocatalytic by themselves [12], such subnetworks cannot be units of selection if they largely depend on the rest of the network.) Thus, network connectivity or the number of species should be finely tuned for the existence of units of selection. On the other hand, the formation of such autocatalytic subnetworks is more feasible in systems based on a specific mechanism of catalysis such as template-directed ligation [46,62,70,71,72].

Here, we further note that not only the network architecture (cores as potential units of selection) but also kinetics, stochasticity, and background reactions affect the existence and stability of multiple compositional states. The stability of such states can be realized by higher-order autocatalysis (i.e., α>1) of core networks; the reaction kinetics leads to frequency-dependent selection among them and which of the multiple stable states one reaches depends on the initial composition.

**Effects of stochasticity and system size**: The size (equivalently, the number of molecules or the volume of compartments such as coacervates) of an autocatalytic system has importance for its evolvability. Even in a system which has only one compositional state in the absence of noise (i.e., when the system size is large and the dynamics are deterministic), stochasticity caused by finite size effects or due to the extinction of some reproductive units (and/or molecular species in an ACS) can lead to the emergence of multiple compositional states [73,74]. For example, in the model of random autocatalytic networks [2,75], if the number of molecules is less than the number of possible chemical species, multiple states can appear.

Additionally, the effects of stochasticity on the reaction system and the probability of transition to novel chemical compositional states depends on the system size. If the system size (*N*) is too large, the intensity of the demographic noise (roughly proportional to $1/\sqrt{N}$) is so small that transition to novel states hardly occurs, precluding any variation for selection to act upon [76,77], as in the case of the transition from an inactive to an active state discussed in the previous subsection.

**Background reactions as a source of variation**: In template replication, errors in replication lead to variation among the resulting sequences. However, replication errors constrain the maximal length of heritable information [11] due to the rapid build up of errors and loss of fidelity, referred to as the “error catastrophe”. In generic autocatalytic networks, background reactions produce variation similar to such replication errors. Background reactions could be non-catalyzed reactions or undesired cross-catalysis between different core networks or between the molecular species within a core. If background reactions are significant, network cores altogether lose their property as units of selection. On the other hand, the appearance of new species by uncatalyzed reactions is important for the creation of and the transition to novel compositional states (e.g., [14,57]). This tradeoff constrains the magnitudes of background reactions in autocatalytic networks as sources of variation and subsequent heredity necessary for evolution (e.g., [72]).

**Selection dynamics of the reproductive units**: Given a source of variation in a self-reproducing chemical system, what conditions exist for Darwinian evolution to occur? As discussed in earlier sections, the molecular reaction kinetics within an ACS affect not only the compositional identity of the core ACS units, $X_{i}$, but also the effective rates, $\lambda_{i}\left(\chi \right)$, at which the core units reproduce. The effect of this reproduction rate, $\lambda_{i}$, can be understood by considering, for example, reproduction dynamics of the form $\frac{d\chi_{i}}{dt}=s\lambda_{i}\chi_{i}^{\alpha }$. Here, α<1 corresponds to sub-exponential growth of the population while α>1 leads to super-exponential growth. When the population growth is sub-exponential, all of the reproductive units survive and coexist in the final state, i.e., there is no selection in the canonical sense. However, if the growth is super-exponential, the prevalence of a particular reproductive unit (and therefore selection) is frequency-dependent and the initial conditions affect the outcome significantly. The most frequent reproductive unit (at the start) has the fastest effective growth rate and thus eventually survives instead of those with the largest reproduction rate $\lambda_{i}$, i.e., there is selection but not Darwinian. Thus, selection among self-reproducing units is fine-tuned, and strictly at α=1, the fittest reproducer is selected, interpreted as selection in the canonical sense.

Furthermore, the evolvability of autocatalytic networks may depend on the assumption of “differential fitness”, i.e., a specific composition can have a function that benefits its growth or is far stabler than the other composition (e.g., see the acquisition of novel functions in Section 3.6 or Section 3.7).

**Multi-level selection**: Can variation and selection be enabled/affected by boundary conditions or compartment dynamics? Selection could occur simultaneously at different levels, i.e., it can act not only on individual (template) molecules or autocatalytic core networks but also on the compartment enclosing the chemical system. Consider competing reproducing autocatalytic units enclosed within compartments (e.g., lipid vesicles or droplets for which growth is coupled to its contents) and that, at an other level, these compartments also compete, as the units of selection among each other. Such multi-level selection has been studied, typically as a stochastic correction mechanism for instabilities such as parasite dynamics (in the hypercycle model) [78,79,80,81]. The size of each compartment and the size of the population (i.e., the number of compartments) are also major determinants for selection [82,83]. In the present context, stochasticity due to smallness of each compartment leads to compositional variation, and each variant can be inherited upon the reproduction of a compartment. (Note: some members of an autocatalytic network can be “keystone species”, and their extinction can cause significant modifications in the network structure [84]. If such species are in a minority, i.e., few in number, fluctuations can give rise to large variation and it has been pointed out that this could be essential for the evolvability of reproducing protocells [85].). Thus, compartmental dynamics enables the variation in compositional states and heredity, with competition among them leading to selection [86].

The evolvability of the autocatalytic network has been debated between critics [65,67] and advocates [2,87]. It was pointed out by Vasas and colleagues [65,67] that autocatalytic networks, specifically based on the GARD model, were not evolvable in the sense that the network compositions were not heritable. However, later work together with Kauffman, which considered another model, did indeed show that non-evolvability is not necessarily true for autocatalytic networks in general [14]—particularly, the conditions for the evolvability of a specific autocatalytic network were identified. Generally, such a disagreement on the evolvability of autocatalytic networks might arise either due to the lack of consideration of several aspects such as the ones we have summarized above (e.g., system size and order of reactions) or due to an insufficient exploration of parameter space.

We end this section with a brief remark on the increase in complexity of an autocatalytic system as a consequence of selection. There have been some initial studies pointing to the role of dissipation in increasing ACS complexity [72]. Lacking a singular view of what the term complexity represents, we note that, for an autocatalytic system, this could mean several things—for instance, increasing complexity could manifest as (i) diversity (network structure or chemical composition) [88,89], (ii) longer (information-richer) polymers constituting the ACS [90,91], and/or (iii) density of catalytic links [8]. (Note: although, we mainly discuss canonical Darwinian evolution, in scenarios involving prebiotic metabolic networks, “evolution” in analogy to the epigenetic adaptation of the present cells has been discussed [92]. For example, more active (higher growth rate) states among multiple (heritable) compositional states are selected even without competition between compartments driven by background reactions and intrinsic noise [72]).

## 3. Experimental Autocatalytic Systems

We now provide an overview of experimental autocatalytic systems and connect them with the theoretical ideas introduced above. Specifically, as we discuss below, experimental systems have not advanced sufficiently to incorporate all aspects of self-reproduction (e.g., compartmentalized ACSs) and canonical Darwinian evolution discussed theoretically. We first present a mechanistic insight into the chemical reaction systems and discuss their autocatalytic nature. Starting with lipid-based systems (Section 3.1), we discuss DNA-based chemistries (Section 3.2), inorganic chemistries (Section 3.3), sugar and small organic-based chemistries (Section 3.4), peptide chemistries (Section 3.5), macrocycles (Section 3.6), and finally RNA-based chemistries (Section 3.7). We then summarize the key aspects of different experimental systems relevant for self-reproduction and Darwinian evolution. We then present a synthesis and identify open questions that remain to be addressed.

### 3.1. Lipid-Based Autocatalytic Systems

A lipid-based autocatalytic system involves self-assembling lipids, where micelles and vesicles self-reproduce via non-covalent assemblies. Luisi et al. [93,94] first demonstrated self-reproducing lipid chemistries where both micelles and reverse micelles catalyze their own production via phase-transfer catalysis which, at the right pH, forms vesicles (Figure 3a). The production of more vesicles increases the hydrophobic environment in the reaction system, further enhancing micellar production via a positive feedback. The reaction system is also autopoietic, viz., the self-reproduction occurs due to autocatalytic reactions within a closed and self-maintained boundary [95,96,97]. Furthermore, autopoetic chiral fatty acid vesicle formation was also demonstrated—though there is no kinetic preference for the enantiomers, homochiral vesicles were found to reproduce at lower temperatures while racemic vesicles were destabilized [98]. Later, following the work of Luisi, Fletcher et al. further developed micelle-mediated autocatalytic reaction systems based on several chemistries involving polar and nonpolar starting materials: thiol-ene reactions, ruthenium-mediated metathesis reactions, and thiol-disulphide exchange reactions [99,100,101].

Fletcher et al. further investigated the hydrophilic-hydrophobic alkene system in which the reactions occur across the biphasic interface to produce amphiphilic products via ruthenium-mediated metathesis reactions (Figure 3b) [102]. The product of the reactions self-assemble into micelles, which enhances the interfacial surface area and consequently increases the rates of reactions in an autocatalytic manner. Among the variety of products formed autocatalytically, using a continuous stirred-tank reactor (CSTR), they further demonstrated the selection of kinetic products, i.e., rate-dependent preferential product formation, wherein the degradation rates were found to be similar [102]. Fitness and selection in such systems can be controlled by the interplay between kinetic and thermodynamic stability of the products. They reported hydrophobicity of the alkenes to be the determining factor for selection in their system [102]. Variation in composition was due to differential experimental protocols, viz., homogeneous phase-separation with and without CSTR. The homogeneous condition resulted in all species having the same relative abundance at different time points (not necessarily a steady-state), whereas dynamical differential compositions were observed for the phase separation and CSTR conditions [102]. This is one of the first claims for the experimental demonstration of selection using lipids. It was shown that, when the system is driven out-of-equilibrium in a CSTR, various compositional (product) states are reached depending on the composition of the flowing food set [102].

In the lipid-based systems, individual micelles or vesicles comprise a unit of reproduction and the composition of the vesicle can, in principle, give rise to variation. However, the variation is a result of the source of substrates and is not de novo. Nevertheless, assemblies of combinations of different chain-lengths of the fatty acids could lead to compositional variation and multiple states. In addition to kinetically controlled selection, such lipid-based systems have the potential for evolution, if inheritable variations can be demonstrated by obtaining varying relative species abundances (chemical composition) through differential seeding with product catalysts [52]. One such rigorous protocol would be to differentially seed the starting food set with products, while it is maintained out-of-equilibrium and to demonstrate more than one compositional steady state. In the work by Fletcher et al. [102], various steady states are obtained by controlled seeding with partitioned food set but not with catalysts.

### 3.2. DNA-Based Chemistries

Kiedrowski et al. developed the first-ever experimental autocatalytic template-based DNA systems [103]. The general scheme is to autocatalytically synthesize templated DNA oligonucleotides starting from chemically modified di- or tri-nucleotides through carbodiimide (EDC)-mediated activation chemistry [103] (Figure 4). Autocatalysis was confirmed by demonstrating linear dependence of the rate of product formation on the amount of template (product of the reaction) [44]. Even though there was an increase in catalysis in the presence of the template, self-reproduction was not exponential due to the inhibitory effect caused by poor release of the newly synthesized product from the template. This was later ameliorated using a “surface-promoted amplification” strategy where the product strand could be washed away, re-generating surface-bound template for the next round of ligation, thereby overcoming the inhibition [104]. Apart from single node catalysis, cross-catalysis has also been demonstrated in a DNA-based autocatalytic system where the product of one reaction node acts as a template for the synthesis of another node [105], constituting a two-member autocatalytic network.

Despite being among the first experimental demonstrations of autocatalysis, DNA-based systems have not been explored extensively; the focus has primarily been on the demonstration of self-reproduction. Notwithstanding the demonstration of cross-catalysis, networks larger than those comprising two nodes have not been created. However, given that the system is based on DNA sequences, which can be designed and synthesized easily, there is indeed the scope to construct diverse networks (including sources of variation) with different numbers of nodes and connectivity. Such developments could lead to the implementation of compositional heredity and selection in such systems.

### 3.3. Inorganic Chemistries

Inorganic chemistries are indeed known to form several complex structures through self-organizing non-equilibrium processes [106]. Owing to their coordination chemistry and complexation properties various self-assembling structures (e.g., nanomaterials) have been synthesized following both top-down and bottom-up approaches [107]. Mineral surfaces have long been known for their catalytic properties [108], owing to which they have been hypothesized to play a role in many inorganic chemistries in the context of the origin of life. These are the so-called clay mineral (montmorillonite silicate) [109], iron-sulphur (surface metabolism) [110], alkaline hydrothermal vent (monosulphide precipitates) [111], and zinc-sulphur [112] hypotheses; experimentally, however, these are not associated with autocatalytic behavior.

A recent study by Miras et al. demonstrated the autocatalytic synthesis of nanosized molybdenum clusters using a stop-flow protocol; the growth kinetics were characterized by a sigmoidal curve [113]. The structure of the autocatalytic set is similar to that of a templated system discussed above where molecular recognition mediated by certain molybdenum clusters act as templates to form other clusters. They further identified the existence of multiple clusters, with implications for variation.

Owing to their simplicity, in comparison with other polymer-based systems, the likelihood of autocatalytic systems based on minerals occurring in early “messy” chemistry is high. In the study of Miras et al. [113], only a few potential products emerge owing to fast kinetics and molecular recognition, from a large possible combinatorial space, implicating selection based on kinetics. However, notions of composition and heredity remain unexplored in these systems.

### 3.4. Sugars and Small Organic Molecule-Based Chemistries

Two different sugar chemistries have been investigated for autocatalytic properties. One is the “triose-ammonia” reaction in which small molecules such as amines and acids are formed upon the reaction of ammonia with triose sugars. These small molecules are known to catalyze many sugar transformation reactions, yielding several products such as amino acids, pyridines, pyrroles, hydroxy acids, furans, pyruvate, and even melanoidin polymers [114]. When a reaction of glyceraldehyde and dihydroxyacetone with ammonium chloride was seeded with a fraction of the reaction product, viz., pyruvaldehyde, the product formation was reported to be ∼2.2 fold faster [114]; this enhancement is a signature of autocatalysis. The triose-ammonia reaction system can synthesize a variety of products, many of which could be catalysts and can lead, in principle, to the emergence of more autocatalytic systems. However, the catalysts neither have been identified specifically nor are the possible catalytic mechanisms elucidated.

Another example is the formose reaction system—aldoses are catalytically formed from formaldehyde under basic conditions in the presence of divalent ions [115,116]. The condensation of two formaldehyde molecules into glycolaldehyde (Figure 5) is catalyzed by higher products (sugars), such as glycolaldehyde, dihydroxyacetone, sorbose, and ribose, thereby enhancing reaction rates and giving an autocatalytic nature to the formose reaction system [117]. Though the formose reaction system is known to result in species of varying chain lengths, chirality, and complexity, the enrichment of any one species is not favored, i.e., it suffers from poor selectivity.

Designing template-based reproduction using small molecules has been a significant challenge. In this regard, Rebek et al. designed self-reproducing imides based on adenine, thymine, and diaminotriazine chemistries [118]. Templating is achieved upon recognition of the product molecule by the monomer substrates through solvent-specific hydrogen bonding. In this reaction system, spontaneous background reactions and product inhibition can be controlled by rational engineering of a spacer between the reaction site and recognition site in the complex [119]. Such principles can also be used to design other efficient reproducers.

Taken together, systems based on small organic molecules are relatively easy to engineer and can possibly be used to construct complex networks. Sugar reaction systems are autocatalytic and have the potential to enrich essential sugars from a prebiotic milieu. Though there exist a few methods to enrich ribose in the formose reaction, there seems to be no demonstration of evolution due to the lack of heredity. Various experimental protocols can be implemented to fine-tune selectivity towards certain specific products. These could involve temperature control, photochemically induced methods, suppression of retro-aldol steps, mineral-based complexification, thiazolium catalyzed oligomerization, stabilization by complexation, and even enantiomeric synthesis (e.g., silicate-based complexification resulted predominantly in tetroses and hexoses, whereas complexification using borate resulted in pentoses) [117,120,121,122,123,124]. Furthermore, side reactions in this system could lead to competing pathways and mechanisms of catalysis that have not been characterized yet. However, such reaction pathways may result in variation required for selection to act upon.

**Figure 5 life-11-00308-f005:**
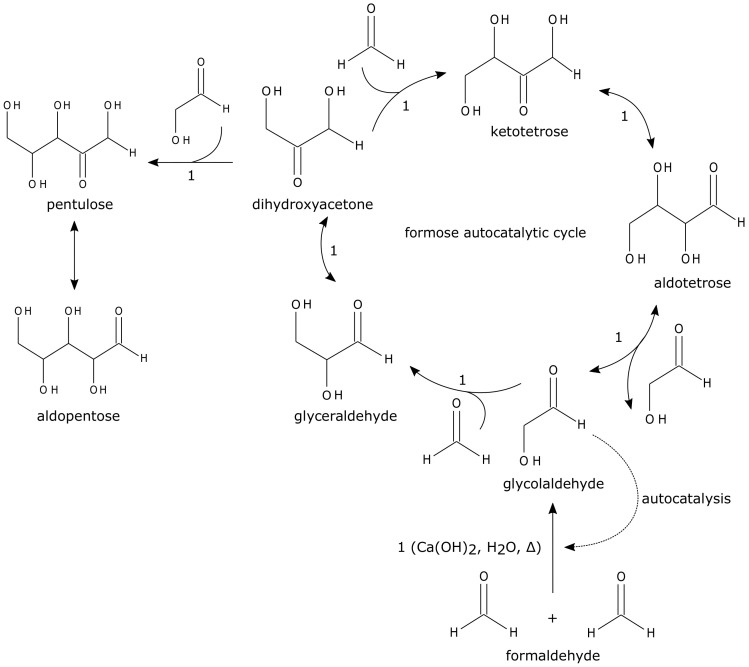
Sugar-based (formose) autocatalytic system. Schematic representation of the autocatalytic system based on the formose reaction. Here, the autocatalytic cycle starts from condensation of two formaldehyde molecules (one carbon each) to form glycolaldehyde (two carbon), which further leads to the formation of higher carbon sugar molecules [115,117]. The reaction is facilitated by basic conditions (**1.** Ca(OH)_2_, H_2_O, and Δ) and product of the reaction (glycolaldehyde). In addition to the autocatalytic cycle, side reactions lead to many higher products (for example, aldopentose) (figure adapted from [123]).

### 3.5. Peptide Chemistries

Both small peptides and nucleic acids are key biomolecules capable of carrying out complex functions. In spite of lacking complementary base-pairing interactions such as in nucleic acids, peptides are still capable of forming higher-order structures, e.g., coiled-coil α-helices. Coiled-coil structures contain 7 amino-acid repeating units (“heptad repeats”, Figure 6a) allowing the peptides to wrap around themselves in a highly stable structure [125]. Using such coiled-coil motifs derived from the “GCN4” yeast transcription factor [126], Lee et al. developed a self-reproducing peptide system [127], where the 32-residue-long motif was broken down into substrate fragments, terminally modified to carry out native chemical ligation to generate another copy of the peptide (Figure 6b). The structure of the autocatalytic system is similar to the one demonstrated by von Kiedrowski using DNA-based chemistries [44]; autocatalytic feedback is measured by carrying out reactions in the presence of different template concentrations. However, as expected due to product inhibition, initial rates of ligation were found to be sublinear (α∼1/2) with template concentration. In a separate study, Lee et al. developed cross-catalytic networks of peptides with two reaction nodes synthesizing each other using a common substrate and another specific one (Figure 6a) [128]. This resulted in a cooperative cross-catalytic network, where both of the reproducers were generated in spite of differences in catalytic efficiency and competition for a common substrate.

Ashkenasy et al. [134] designed 81 peptide sequences in silico (by varying “g” and “e” residues in the heptad repeats (Figure 6a)) to predict potential self/cross-catalytic nodes and catalytic links in the network based on the stabilities of substrate-template complexes [135]. This allowed them to predict 25 nodes connected by 53 catalytic links, several of which were also experimentally realized. While this demonstrates that an understanding of the molecular properties can enable designing complex networks, modifying interacting residues (“a” in heptad repeats) can also allow the construction of diverse nodes as this position controls a specific recognition between peptides.

In parallel to the work above, Yao et al. reported coiled-coil helices of tropomyosin capable of self and cross-reproduction [136,137]. By modulating the “e” and “g” positions of the heptad repeat (Figure 6a), they demonstrated the control of self-reproduction by modulating pH [138] (when “e”/“g” contains glutamic acid residues) or ionic strength [139] (when “e”/“g” contains lysine residues), allowing the creation of a network in which catalytic links are modulated by the environment. In order to improve autocatalytic efficiencies and to reduce product inhibition, these peptides are shortened [140] or modified to contain a proline-kink at the “d” or “e” positions in the heptad [141].

While the systems discussed above use irreversible native chemical ligation, Dadon et al. employed a reversible trans-thioesterification chemistry to construct template-based self-reproducing peptides [142,143,144]. Here, similar coiled-coil architectures are used; however, the nucleophilic fragment is terminally modified to contain a thiol (using thiol glycolic acid) such that they can form reversible thioester bonds, since they are susceptible to free thiols, readily decomposing to starting material. This reversibility (decomposition) correlates well with the stability of the substrate-template complex, and therefore, the stable complexes result in efficient reproduction [143]. As discussed earlier in Section 2.4, multiple stable steady states are critical for Darwinian evolution in self-reproducing networks [16,55]. This system was then further exploited to experimentally demonstrate bistability in a chemical network due to the higher-order catalysis by a dimer template [129]. Since there is a difference in the rate of formation of thioester product (T) and its decomposition to substrates (E and N), in the presence of thiols, the reaction starting with only substrates reaches different distributions of E, N, and T (low steady state) compared to when the reaction starts with T + thiol (high steady state) [129] (Figure 6c). The bistable behavior is further characterized by keeping the complete reaction system out of equilibrium using a reducing agent TCEP (tris(2-carboxyethyl) phosphine hydrochloride, which reduces disulphides to thiols) as fuel [145] (Figure 6c).

Self-reproducing peptide systems possess several features required for evolution to occur. Though they do not have inherent catalytic properties and diversity (e.g., sequence diversity of nucleic acid catalysts), templated-ligation using coiled-coil structures in the peptide-system results in exponential autocatalytic growth. Structurally too, all the self-reproducing peptides discussed here are based on the same coiled-coil heptad repeat structure with little flexibility on the amino acid requirements. In spite of this, complex cross-catalytic networks with as many as 25 nodes can be formed [134]. In this system, a recombination-like mechanism, employing reversibility which allows new combinations of reproducer to emerge from the peptide pool, can be a source of variation. Peptide-based reproducing systems are also robust against molecular perturbations (chemical changes) as native sequences dominate a reproducer pool even in the presence of mutant sequences [146]. Furthermore, peptide-based systems are the only ones where bistability has been demonstrated experimentally [129]; however, since bistability is between two concentrations of the same reaction node, the notion of distinct chemical compositional identity has not yet been established. Another important feature of the self-reproducing peptide system is the ability to control the autocatalytic and spontaneous reactions using environmental control such as the pH and salt concentration [138,139]. Such properties allow environmental control of the switch between autocatalytic and non-autocatalytic pathways.

### 3.6. Macrocycle-Based Systems

Another important chemical system that has the tendency to self-assemble (in long chains; fibrils) is closed-ring structures formed by macrocyclic compounds [147]. Exploring this feature, autocatalytic reaction systems based on peptide-containing macrocycles have also been developed. Exploiting Dynamic Combinatorial Libraries (DCLs [148]), Otto et al. isolated self-reproducing macrocycles starting with peptide-based substrates [130]. These substrates contain thiol groups, which in the presence of oxygen from the environment form disulphide bonds with each other, resulting in mixtures of macrocycles containing trimers to octamers of the peptide containing substrates (Figure 6d). Self-reproduction is then driven by the self-assembly of these molecules into secondary structures (fibril formation aided by beta-sheets of peptide chains) with their subsequent fragmentation (Figure 6d) [132]. Autocatalytic growth is further confirmed through seeding experiments where adding a macrocycle of a specific ring size at the start of the reactions enriches its own production [133]. This system is further used to demonstrate a form of “proto-metabolism”, where the reproduction of macrocycles is coupled to a metabolic reaction that feeds back to the self-reproduction process [149]. Here, the cleavage of *Fmoc* (a protection group [150]) from a substrate molecule was found to be accelerated in the presence of macrocyles, in turn generating alkene as a product [149]. Alkene aids the oxidation of thiols, therefore enhancing the reproduction of the macrocycle by forming more of the disulphide-containing trimers/tetramers required as substrates for the assembly of a hexamer. It is a pioneering demonstration of a self-reproducing system acquiring new metabolic functions feeding back to its own reproduction. However, whether such activity is emergent (e.g., from a DCL) remains an open question.

Non-covalent assemblies of macrocycles show self-reproduction and are capable of forming different compositions [151,152], but a clear demonstration of networks is lacking, in particular, a lack of multiple catalytic nodes and kinetic rates (edge weights). The combinatorial nature of DCLs can be exploited to construct large compositional landscapes, which under appropriate selection pressures, could lead to heritable compositions.

### 3.7. RNA Chemistries

The autocatalytic assembly of molecules from RNA fragments has been reported in two separate systems, in which assembly is driven either by (i) a template-based ligation reaction or (ii) energy-neutral recombination reactions.

The first system is developed from an in vitro ligase ribozyme [153,154], which catalyzes the formation of the 3′–5′ phosphodiester bond between substrate RNA fragments and the 3′-end of a ribozyme (“R3C ligase”) via specific base-pairing recognition. The “R3C” ligase is engineered to function as a self-reproducing system [155] using the same strategy as DNA-based ones (A+B⟶T, where A and B ligate over the template). A symmetric dimer of the “R3C” ligase is constructed by duplicating essential structural elements of the ribozyme, comprising of two halves (A and B). These parts can then be templated as substrate fragments (food set) for the catalytic production of a new copy of the ribozyme (Figure 7a). By manipulating the base-pairing recognition while keeping the catalytic structural element the same, cross-catalytic reproduction (instead of self-reproduction) of two ribozymes is achieved (Figure 7a) [156,157].

Later on, starting with a substrate pool of reproducers (48 substrates for 12 pairs of cross-reproducing ligase ribozyme), Lincoln and Joyce [156] implemented an out-of-equilibrium serial-transfer protocol to select for the fastest reproducing species. Surprisingly, they found a recombinant product (“A5B3”, combination between substrates differing from the expected types) to be the fastest reproducing species—this may be interpreted as the selection of a de novo species. However, while this species is the fastest reproducer in the presence of all 48 substrates, it is not so (i.e., it is a poorer reproducer) when compared with the parent reproducing species in isolation. This suggests the emergence of a connected network of cooperative cross-reproducers, enabling the collective reproduction of the de novo recombinant species. However, the unambiguous identification of such a network requires mapping of all the catalytic links and measuring their relative strengths.

Another RNA-based chemical system capable of autocatalytic self-reproduction is derived from the group I intron of the *Azoarcus* bacterium [158]. This ribozyme is engineered by breaking it into small fragments that can self-assemble in an autocatalytic manner to generate functional ribozymes [159]. The ribozyme can be broken at multiple (1 to 4) positions, resulting in inactive substrate fragments (food set), viz., W, X, Y, and Z (Figure 7b). Recognition, i.e., specificity in this system, is dictated by the base-pairing between a 3-nucleotide-long Internal Guide Sequence (IGS) at the 5’-end of the ribozyme with the cognate 3 nucleotide stretch at the 3’-end of the substrate fragments tag [43]. When mixed together, the RNA fragments rapidly hybridize into non-covalent complexes [W:X:Y:Z] and are catalytically active. Subsequently, the non-covalent junctions are joined by covalent bonds. This reaction is catalyzed by both the non-covalent and the fully formed (covalent) ribozymes (Figure 7b). Taken together, this is the autocatalytic self-reproduction of the ribozyme, since the product of the reaction feeds back to its own synthesis via specific recognition of the cognate tag [159]. Later, Vaidya et al. modulated the recognition between IGS and the cognate tag (by changing the middle nucleotide, “GMG”/“CNU”, where M and N can be any of the four nucleotides) and thus constructed cross-catalytic reactions [53]. This allowed for the generation of a network comprising of 48 (16 IGS-tag pairs for 3 different junctions) different WXYZ catalysts, and the roles of cooperativity and selfishness in such autocatalytic reaction networks were studied. They also subjected these 48 substrates to a serial-transfer experiment, demonstrating the emergence of cooperative networks of reproducers (cross-reproducers) over selfish self-reproducers. Later, Yeates et al. specifically measured the autocatalytic rate constants for all 16 IGS-tag reaction pairs [161] and found that reactions catalyzed by the recognition of Watson–Crick pairing between the IGS and cognate tag were higher than the corresponding ones for nonspecific pairing.

In order to establish the role of network connectivity in controlling compositional variation, growth, and robustness, Ameta et al. [160] constructed multiple diverse networks (an ensemble of >20,000 isolated network configurations) using combinations of the food set. Employing a high-throughput strategy by combining droplet-based microfluidics and barcoded-sequencing, they quantified the compositional landscape (relative proportion of ribozymes in a network) of the networks at an unprecedented resolution. Studying this network landscape, it was observed that network topology plays a crucial role in dictating network growth. At the same time, increased connectivity buffers the networks against compositional variation (i.e., due to the random addition or removal of new species). However, there is a tradeoff between the network variation and robustness. This arises since variations can still occur when a weakly connected node of the network is targetted with a strong catalytic link, sustaining the interplay between growth and compositional robustness (Figure 7d). Similar to macrocycles [149], autocatalytic reaction networks based on the *Azoarcus* system also displayed “proto-metabolism”—specifically, self-reproduction is maintained by the catabolism of modified RNA fragments into native substrates [162]. Such emergent metabolic behavior, apart from providing robustness, demonstrates that self-reproducing RNA systems are also able to acquire novel functions.

Autocatalytic reaction networks based on RNAs have the potential to exhibit Darwinian evolution. Using two different strategies, templated-ligation, and recombination, reproducers capable of self- as well as cross-catalysis have been developed. Cross-catalysis is well-exploited for the *Azoarcus* system to construct diverse networks to explore the underlying mechanisms governing growth and robustness in such networks. In contrast, only two-membered networks, where a pair of reproducers catalyze each other’s synthesis, were explored using templated-ligation systems. However, the emergence of a recombinant ligation-based reproducer in the experiment by Joyce et al. [156] suggests the presence of an underlying and well-connected network. While both RNA-based systems can self-reproduce and exhibit variation (though limited, this can be achieved by recombination reactions in the case of the *Azoarcus* system), Darwinian evolution in such networks also requires the demonstration of compositional heredity and selection. Compositional heredity was attempted in a test network from the *Azoarcus* system but could not be faithfully demonstrated because, due to spontaneous (background) reactions, different compositions converge to a single state after two steps in a serial-transfer experiment [160].

### 3.8. Summary of Experimental Systems

Table 1 summarizes the different experimental systems, described in previous sections, that can form self-sustaining reaction networks from a diversity of chemistries. As discussed earlier, a unit of selection is a self-reproducing unit with an identity and can either be comprised of a single chemical species (e.g., templated polymer) or an autocatalytic network comprised of multiple chemical species (having a single compositional identity). It is not clear if autocatalytic networks can be constructed for all the experimental systems described above (e.g., lipid-based systems and inorganic molybdenum self-reproducing clusters), and we focus our discussion here on systems in which the construction of autocatalytic reaction networks has been demonstrated. In spite of the possibility for cross-catalysis, network construction has not yet been demonstrated in some systems (e.g., sugars and macrocycles). Self-reproducing autocatalytic networks have been constructed using DNA, RNA, and peptide chemistries; however, networks comprising of a large (>2) number of nodes has been demonstrated only for RNA- and peptide-based systems. Steady-state compositional identity of the autocatalytic networks has so far only been established for *Azoarcus* RNA-based systems [161,163]. However, the compositional identity can be affected by the presence of spontaneous reactions, i.e., background reactions (or due to catalysis by non-autocatalytic pathways) and degradation. In the *Azoarcus* system, a spontaneous non-covalent (but weakly catalytically active) assembly of the food set can cause such background reactions. In addition, nonspecific hydrolysis in the RNA-based system can result in degradation of the chemical species.

Experimental demonstration of the compositional heredity of an autocatalytic network remains an open challenge. This not only is an experimental challenge but also requires the application of theoretical ideas for the identification of suitable network architectures, appropriate dynamical regimes, and the concomitant development of suitable protocols (boundary conditions). A fundamental requirement is the existence of multiple out-of-equilibrium heritable states, i.e., multi-stability. An experimental system should therefore exhibit multiple stable compositions that result from the same food set. While the existence of bistable (two states) compositional identities has been demonstrated using peptide chemistries [129,145], such multi-stability has not yet been demonstrated in other experimental systems.

Robust compositional variations of the self-reproducing unit should arise via de novo mechanisms rather than trivially be imposed by the boundary (initial) conditions such as in the food set composition. However, so far, there is no unambiguous experimental demonstration of the generation of such variation. (Note: this may indeed be possible in chemical systems capable of “recombination” (e.g., *Azoarcus* recombinase ribozyme) since the shuffling of different (random) fragments (units) generates new chemical species. Such new combinations may have fitness consequences as observed in the ligase-based self-reproducing RNA system where a recombinant product emerged as the fastest reproducer [156].)

Variation in an autocatalytic network can also occur due to the spontaneous (random) addition or removal of species from the network. Such a scenario has been explored experimentally using the *Azoarcus* system where a compositional landscape of thousands of networks has been used to understand tradeoffs between compositional variation (which requires low network connectivity) and robustness against addition or deletion of new species in network (which requires high network connectivity) [160].

Briefly, we also note that the adaptation of high-throughput analytical techniques are crucial in the development of experiments. This is particularly so, given that many of the experimental chemical systems cannot be subject to high-throughput analysis, limiting their scope for the exploration of multiple parameter regimes and protocols. Chemistries based on small molecules need mass-spectrometric methods, which typically have a lower throughput compared to sequencing methods employed [53,160,164,165] for nucleic acid-based self-reproducing systems. Furthermore, droplet-based microfluidics can be combined with next-generation sequencing, massively increasing the exploration of multiple parameters [160]. Finally, spectroscopic approaches such as florescence assays for real-time analysis, as was done for the ligase-based self-reproducing system [166], can expand the experimental possibilities.

## 4. Future Perspectives

The realization of Darwinian evolution in minimal chemical systems and the quest for minimalistic life-like systems will require intensive investigations of actual network topologies, energetics, and kinetics to bridge theoretical models with experimental possibilities. In the context of the origins-of-life scenario, autocatalytic systems as the first unit of life should (i) spontaneously emerge from an abiotic chemical mixture (which could include catalytic and non-catalytic building blocks) and (ii) be evolvable. Although there are theoretical studies that discuss the conditions for such an emergence, there are no experimental demonstrations so far. For both theoretical and experimental studies, elucidating the conditions for spontaneous emergence and evolvability of autocatalytic systems is imperative. Experimental demonstration of the emergence of ACSs from a prebiotic soup or from a pool of random chemical moieties (e.g., a library of oligonucleotides) requires the development of robust and sensitive enrichment protocols (such as SELEX (Systematic evolution of ligands by exponential enrichment) [167,168]), which should be able to select the complete autocatalytic network from the pool. However, there is currently no protocol established for the in vitro selection of collectively emerging reproducing units (core autocatalytic network cycles) from randomized pools of chemicals. Here, it is also important to consider the diversity of the initial random pool as it is hinted that it plays a critical role in the emergence of an ACS from the pool [21,169,170]—designs of such a random pool should particularly take into account limitations on catalytic density, reaction kinetics, and network architectures, as we have discussed earlier from theoretical considerations.

Furthermore, the possibility of emergence cannot be discussed in isolation from the evolvability of the system. For example, large non-catalytic background reactions (or large connectivity between networks) makes it favorable for the spontaneous emergence of autocatalytic self-reproducing systems; however, this could affect the ability to inherit information (and evolvability) of the system. Indeed, the magnitude of background reactions are different among the various autocatalytic systems reviewed in the previous section (e.g., the background reaction rates for the *Azoarcus*). Therefore, there might be an optimal background reaction rate that could be suitably engineered by environmental changes to suit the particular stage of chemical evolution.

Demonstration of the heredity of autocatalytic reaction network compositional identity and subsequent competing interactions leading to selection is currently lacking. This requires both theoretical and experimental advances: indeed, recent theoretical work (Matsubara et al., unpublished) has identified suitable network architectures and appropriate dynamical regimes required for heredity of the compositional identity of an autocatalytic network; the system based on the *Azoarcus* ribozyme, among others, is a promising experimental candidate to validate these theoretical ideas. Furthermore, while there are some promising hints for the possibility of variation arising in the different experimental systems we have discussed, the demonstration of de novo variation and its coupling to the self-reproducing and evolutionary dynamics remains an open challenge. Chemistries in which recombination can occur are rather straightforward experimental candidates for such explorations. Other subtler sources for the generation of variation, such as spontaneous background and degradation mechanisms, will require a combination of theoretical and experimental advances for progress, specifically in terms of network architectures and kinetics.

Can the collective dynamics of ACSs give rise to templated replication? A hint comes from contemporary biological systems. A “replicase” (RNA-based RNA polymerase) is a molecule that mediates copying of template sequence information, and the emergence of a similar mechanism/structure could be critical for the origin of templated replication; a candidate for such a replicase is an in vitro selected ribozyme, which can copy up to 200 nucleotides [171,172,173]. Initial attempts to create such replicases involved group I introns, such as the *Tetrahymena* ribozyme [174] and RNA ligases [154]. In the case of the group I intron, both single nucleotides and fragments were shown to ligate over the template using a trans-esterification reaction. Such results indeed hint that an ACS system based on the catalytic activities of introns (e.g., *Azoarcus* ribozyme) and ligases might give rise to a “replicase”, leading to the advent of a template-based replication system. Moreover, even the model “replicase” discussed above was shown to emerge as a component of networks of fragments interacting with each other [175].

While templated replication is already reminiscent of present-day living systems, our discussion thus far has not considered a central role for compartments that could indeed be critical at every step of the reproductive and evolutionary dynamics [13]. We end with some brief remarks on physicochemical constraints and compartmentalization—aspects that are not discussed at length in our review but are important for the emergence of life.

The likelihood and emergence of autocatalytic sets are, in general, constrained by historical geochemical conditions and physicochemical properties. Such geochemical constraints could broadly be categorized as (i) availability of food molecules and (ii) reaction conditions, i.e., temperature and concentration, whereas physicochemical constraints could be related to (i) the rate of spontaneous reactions and (ii) the probability for any molecule to be a catalyst. As such, very few experimental studies such as the ones we discussed in Section 3 take such constraints into consideration. These experiments and the models discussed in Section 2 assume an abundance of basic building blocks, i.e., monomers such as amino acids or (activated) nucleotides. However, the synthesis of such building blocks in prebiotic conditions has been investigated experimentally (e.g., [176,177,178]). In addition, the spontaneous polymerization of amino acids and nucleotides have been attempted in conditions mimicking the tidal pool (i.e., wet–dry cycles) [179], thermal vents [180], or mineral assistance [181].

Theoretically, a deep investigation into the physicochemical and geochemical properties of autocatalytic polymers will elaborate the assumptions on models of chemical reaction networks [59,182] (e.g., kinetic parameters) and enable a more precise discussion on the probability of the spontaneous emergence process [6,60]. For example, the probability *p* for random peptides to be enzymes is estimated as $p∼10^{-5}$ indirectly from the random antigen-antibody recognition probability [20] (and similar alternative estimations in [170]); however, such an estimation could be too optimistic and may require the exploration of parameter space based on the constraints discussed here. Readers may find other studies or reviews [183,184,185,186] where such conditions are discussed in greater detail.

Compartmental dynamics are a natural way in which autocatalytic systems can couple with sustained self-reproduction via growth and division of compartments. In addition, compartmental dynamics—with an appropriate volume (system size) as discussed theoretically—could be important for emergent evolutionary dynamics such as multi-level selection. Compartmentalization can be important for the protection of a chemical species from degradation, thus providing robustness. Various model systems of compartments such as vesicles, droplets, coacervates, and their dynamics have been studied [187,188,189,190]. Recent theoretical work suggested that phase separation dynamics driven by coupled reactions could give rise to compartments that can grow and divide [191]; however, experimental demonstration remains an open challenge.

## 5. Glossary of Terms

**(Canonical) Darwinian evolution:** The dynamics of a heritable variation being selected from among a population of reproducing units according to differential fitness. We define evolution more specifically in Section 2.4 in dynamical systems terminology, and we refer to it as Darwinian evolution in a “canonical” sense.**Self-reproduction/replication:** The self generation of a new copy of a unit/individual. In the present context, self-reproduction of a chemical system is realized by autocatalytic reactions. Although reproduction and replication are regarded as conceptually similar in general, in the present review, we only use “self-replication” for a generation of new copy via template-directed replication.**Template-directed replication:** A chemical reaction process where a sequence of a polymer is copied monomer-by-monomer, directed by another polymer as a template. The replication of DNA plays a major role in the present life to propagate information across generations.**Metabolism-first scenarios:** An origins-of-life hypothesis that surmises that metabolism proceeded template-directed replication, e.g., Oparin’s coacervate hypothesis (its counterpart are replication-first scenarios).**(Chemical) composition:** The abundance (or the number) of each species in a chemical system.**Compositional heredity:** The propagation of information via chemical composition, without template-directed replication (alternatively named as “composome”).**Chemical reaction network:** A network composed of chemical species (nodes) and reactions among them (edges); refer to the main text for a more precise meaning of “edges”.**Dynamical system:** A system describing the time dependence of variables (in the present context, chemical composition).**Stochastic process:** A process describing the time course of random variables (in the present context, chemical composition with random fluctuations). The dynamics of the chemical composition should be treated as a stochastic process when the system size (i.e., the number of molecules) is small.**Catalyst:** A molecule that accelerates the rate of a chemical reaction; the process is termed as catalysis. In particular, a catalytic protein is called an enzyme and a catalytic RNA is called a ribozyme.**Autocatalytic reaction:** A chemical reaction in which the catalytic molecules catalyze their own production; the process is called autocatalysis.**Background reaction:** In general, a chemical reaction that is not associated with autocatalytic reactions (or autocatalytic sets). Note that background reactions are context-dependent.**Autocatalytic sets/cycle (ACS):** A set (or cycle) of reactions such that the catalyst of the reactions is synthesized from the food molecules by a series of reactions in the set. There are several formal definitions for the ACS (e.g., RAF (Reflexively Autocatalytic and Food-generated) set; for details, refer to the main text Section 2.1).**Food molecules (set):** A molecule or a set of molecules that is/are externally supplied as substrates to an autocatalytic system.**Network percolation:** A transition from a network with disconnected clusters of nodes into a network with almost all nodes connected to each other due to the increase in the number of edges in a random network (graph).**Continuous Stirred-Tank Reactor (CSTR):** A boundary condition of a reaction system where food molecules are continuously supplied while all molecules continuously diffuse out (typically known as a chemostat in the biological context).**Bi/multi-stability:** A property of a dynamical system with multiple steady states. A large enough perturbation can cause transitions among such states.**Dissipative structure:** An emerging stable structure in a system with open boundary conditions, for which the characteristics have been studied in non-equilibrium physics since the pioneering works by Ilya Prigogine (also known as dynamic kinetic stability in systems chemistry).**Quasi-species model:** A model, introduced by Eigen, describing the population dynamics of replicative template (nucleotide) polymers.**Hypercycle:** A cooperative structure between template polymers proposed by Eigen, where the replication of template polymer is catalyzed by the another polymer.**(Spontaneous) emergence of ACS:** A transition process where an ACS first emerges from the food molecules (or “messy chemistry”), mediated by background reactions and stochastic fluctuations.**Units of selection:** Units that can be subjects of Darwinian evolution; according to the formulation by Lewontin, the necessary conditions for the population of such units are (i) variation, (ii) heredity, and (iii) differential fitness.**(Autocatalytic) core network:** A subset of the autocatalytic network, which is autocatalytic in itself, and independent from the rest of the network. Core networks could be units of selection and are hence significant for the evolvability of ACS.**Amphiphile:** A chemical compound with both hydrophilic (affinity for water) and hydrophobic (affinity for lipids or fats) properties.**Micelle/Vesicle:** Micelles are spontaneous spherical supramolecular arrangements of amphiphiles in aqueous solutions, formed due to amphiphilic interactions. Vesicles are similar supramolecular structures comprising of fluid enclosed by a bilayer of amphipathic molecules.**Autopoiesis:** An autocatalytic reaction that takes place within a closed boundary and contributes to the sustenance of the boundary. An illustrative example of an autopoietic system is the self-reproduction of lipid micelles/vesicles (see Section 3.1).**Stopped-flow:** A flow mixing setup used to investigate the kinetics of very fast reactions.**Molybdenum clusters:** Neutral or charged chemical compounds comprising of many metal (molybdenum) moieties, usually with significant metal–metal interactions.**Dynamic Combinatorial Library (DCL):** A library (or set) of simple chemical species (building blocks), from which a combinatorial variety of product molecules can be generated via reactions between the building blocks.**Macrocycle:** Chemical compounds with closed ring-like structures of 12 or more members.**Oligomers/oligonucleotides:** Short strands of polymers (for example, a 10-mer is an oligomer that is ten monomers long). In particular, oligomers of DNA or RNA molecules are called oligonucleotides.**Coiled-coil:** Protein structural elements composed of alpha helices (>2) wrapped around together to form spiral structures.**Ligation:** Reactions involving covalent bond formation between two chemical species.**Cross-catalysis:** The process (chemical reaction) in which a molecule (catalyst) catalyzes the formation of a different molecule.**Recombination:** A reaction where polymers (DNA, RNA, or peptides) are cut and then re-joined/spliced together to generate new combinations.**Polymerization:** Reactions in which small molecules (monomers) are covalently linked to produce longer molecules (polymers).**Seeding:** A protocol of adding a molecule to a chemical reaction system in a concentration-dependent manner.

## Figures and Tables

**Figure 2 life-11-00308-f002:**
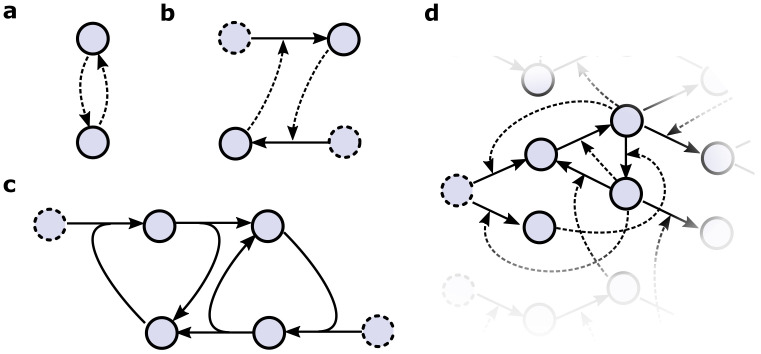
Several illustrative autocatalytic networks with different definitions. Each node represents chemical species (reactant and molecule), but the meaning of edges differs among them: (**a**) edges represent catalysis of the production of the other molecules. Food species are omitted. (**b**) Solid edges represent the reaction (change of stoichiometry), which is catalyzed by another; dashed edge denotes their catalysis. The food species is shown as a dashed node. (**c**) Edges represent the change in stoichiometry (but not catalyzed by another member explicitly). (**d**) A (part of a) randomly generated autocatalytic network, with the same meaning of nodes and edges as in (**b**).

**Figure 3 life-11-00308-f003:**
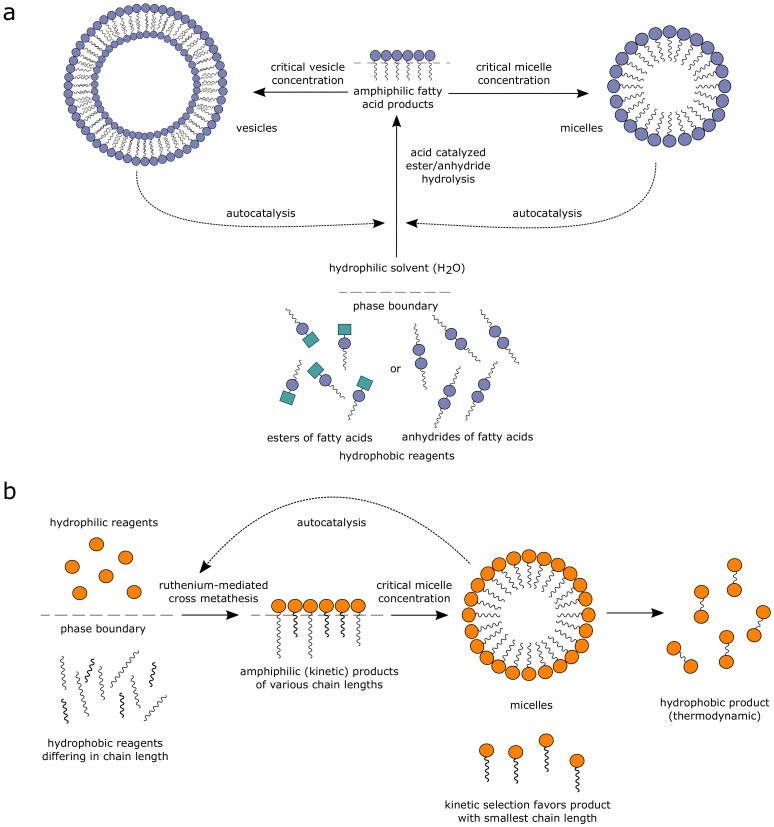
Lipid-based autocatalytic systems. (**a**) Schematic representation of the self-assembly of ester or anhydrides of fatty acid molecules into micelles. Here, micelles catalyze the production of more micelles by enhancing the solubility of the substrates in aqueous phase [94]. Such a process can also lead to the formation of vesicles from the heterogenous mixture of protonated and non-protonated fatty-acid molecules. Similar to micelles, these vesicles can also lead to the production of more vesicles, making the process autocatalytic [95]. (**b**) Another self-assembly system based on lipids in which amphiphilic substrates are generated in situ, employing metathesis reactions between hydrophobic and hydrophilic alkenes. Similar to (**a**), these substrates assemble to form micelles in an autocatalytic manner, but in this case, the starting material is a mixture of heterogenous substrates of varying chain-lengths [102]. This system demonstrates the enrichment of only one type of amphiphile through miceller autucatalysis, raising the possibility of selection in the lipid-based self-reproducing system (figure adapted from [102]).

**Figure 4 life-11-00308-f004:**
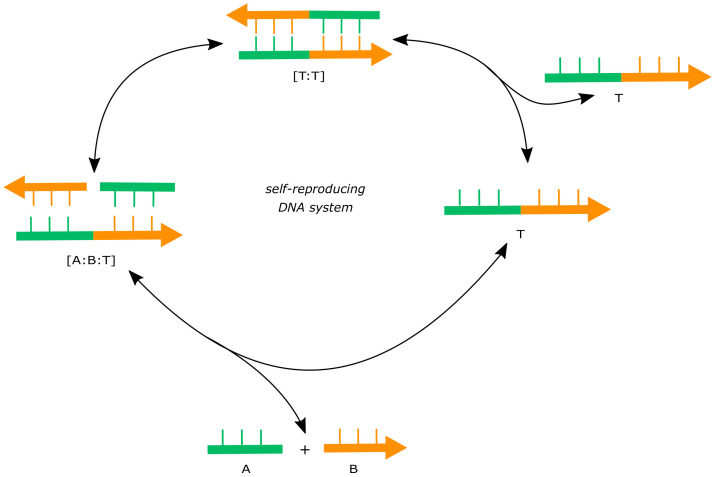
DNA-based autocatalytic system. Schematic showing a self-reproducing system based on DNA oligonucleotides. Here, two chemically activated DNA fragments (A and B) hybridize over the template molecule (T), leading to the formation of an [A:B:T] complex [103]. This leads to templated ligation of the substrates forming a [T:T] complex, generating a complementary copy of the template (T) in an autocatalytic manner [103] (figure adapted from [103]).

**Figure 6 life-11-00308-f006:**
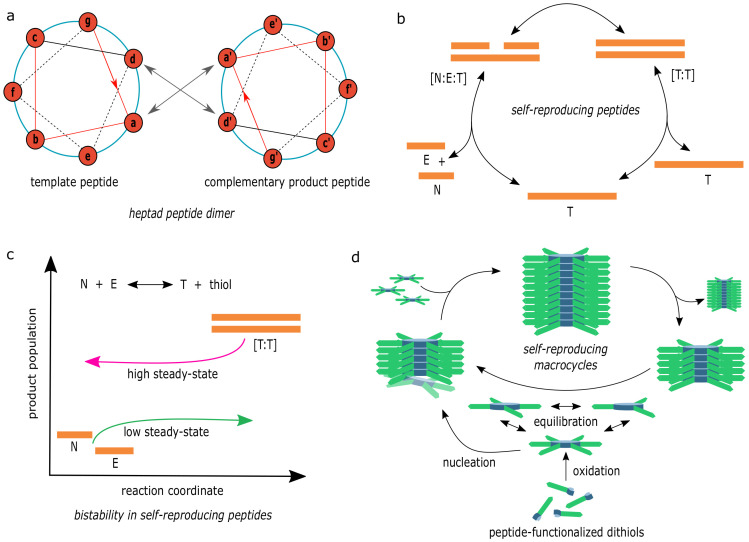
Peptide- and macrocycle-based autocatalytic systems. (**a**) Schematic representation of the heptad structure of coiled-coil α-helix and the amino acid positions important for the interactions between two peptide chains. Here, amino acids at the a, a’ and d, d’ positions are involved in recognition [127,128] (figure adapted from [127]). (**b**) Self-reproducing peptide system based on the heptad structure shown in (**a**). Here, fragments from one of the peptides are chemically activated to act as substrates (E and N) over the template (T) peptide to generate the [N:E:T] complex. This facilitates the ligation of substrate fragments forming the [T:T] complex, generating the complementary product in an autocatalytic manner [127] (figure adapted from [127]). (**c**) Bistability in the peptide-based autocatalytic system. Here, the reaction starting with a low amount of T and high amount of E, N reaches a different steady state (low steady-state) compared to when it started with higher amount of T and thiol (higher steady-state) [129] (figure adapted from [129]). (**d**) Schematic representation of the self-reproducing macrocycle system based on peptide-functionalized dithiol chemistries. Here, the dithiol substrates are oxidized to form macrocycles of varying sizes through disulphide bonds between multiple substrate molecules (for example, three, four, or six membered rings) in a Dynamic Combinatorial Library (DCL) setup [130,131]. Nucleating with one of the macrocycles (for example, six membered macrocycle) leads to the formation of stacks through interactions mediated by the peptide backbones, further leading to the elongation and growth of these rod-shaped materials [132,133]. Mechanically agitating the mixture leads to fragmentation of the growing material, and the fragments in turn act as seeds for self-assembly of more stacks in an autocatalytic manner [132,133] (figure adapted from [130]).

**Figure 7 life-11-00308-f007:**
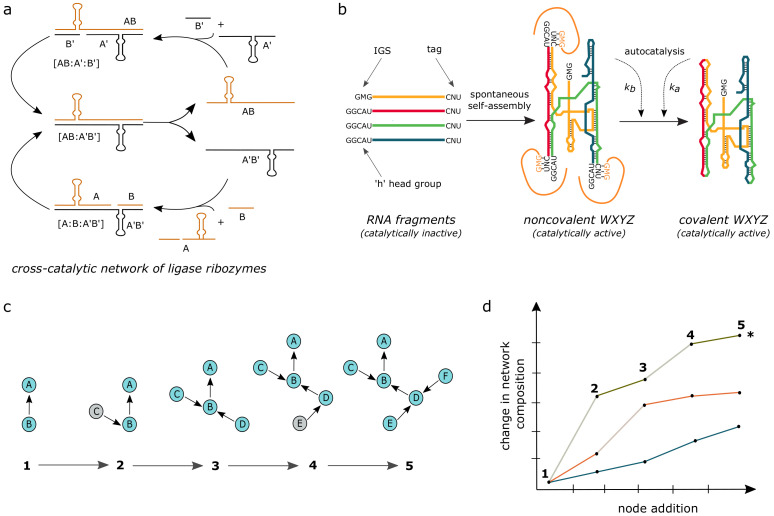
RNA-based autocatalytic systems. (**a**) Schematic representation of the cross-catalytic network [156] developed from the RNA ligase ribozyme [153,154] (figure adapted from [156]). (**b**) Autocatalytic RNA system based on group I intron ribozyme from *Azoarcus* [158]. The fragments of *Azoarcus* ribozyme (W, X, Y, and Z) self-assemble of form non-covalent complex [W:X:Y:Z], which is converted into covalent ribozyme (WXYZ) in an autocatalytic manner [159]. Here, both covalent as well as non-covalent versions of ribozyme can act as catalysts [43] (figure adapted from [159]). (**c**) Schematic of an autocatalytic network, where nodes are added sequentially. (**d**) Compositional variations in the network (Figure 7c) over the trajectories where [160], starting from a two-membered network (1), nodes are added and compositional variation is measured (as the change in composition before and after adding a node). Strong variation arises (grey in the curve with [*****]) when isolated nodes are targeted with a strong catalytic link [160]. These networks can have multiple such variation events based on the network topology and the catalytic strength of the incoming node. Examples of no variation (blue curve), single variation events (orange curve), and double variation events (green curve with [*****]) are shown here (figure adapted from [160]).

**Table 1 life-11-00308-t001:** Various autocatalytic chemistries and their evolutionary properties.

Chemical System	Reproducing Unit	Networks	Boundary Conditions	Variation	Heredity
Lipid-based [94,102]	Chemical composition ^1^	N/A	CSTR	Food set-generated ^2^	Protocol-mediated ^3^
DNA-based [103,105]	Oligonucleotides	1 network, 2 nodes [105]	Equilibrium	Reaction kinetics ^4^	N/A
Inorganic-based [113]	Molybdenum clusters	N/A	Stopped-flow	N/A	N/A
Sugar-based [115,116]	Sugars (C2–C5)	N/A	CSTR [116]	Reaction kinetics ^4^	N/A
Peptide-based [127,134,138]	Peptides	1 network, >20 nodes [134]	Both ^5^	Reaction kinetics ^4^	Concentration-mediated ^6^
Macrocycle-based [148]	Macrocycle assemblies	N/A	DCL	Reaction kinetics ^4^	N/A
RNA-based [155,156,159]	Chemical composition ^1^	>20,000 networks [160], >40 nodes [53]	Both ^5^	Reaction kinetics ^4^	Differential seeding ^7^

^1^ Chemical composition: abundance (or number) of product molecules in micelles, vesicles [2], or networks of RNA [160]. ^2^ Food set-generated: variation arises due to the composition of food set flowed under continuous stirred-tank reactor (CSTR) conditions [102]. ^3^ Protocol-mediated: differential compositional states are obtained due to different experimental protocols, namely equilibrium, phase separation, and CSTR conditions [102]. ^4^ Reaction kinetics: variation owing to a combination of background reactions and boundary conditions. ^5^ Both: equilibrium [129,160], and out-of-equilibrium conditions [53,145]. ^6^ Concentration-mediated: bistable steady states are obtained by starting with different concentrations of reactants and products [129] or by changing the concentration via a chemical fuel [145]. ^7^ Differential seeding: multiple steady states are obtained by controlled seeding with different RNA catalysts [160].

## Data Availability

Not applicable.

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
