# Peer review of "Self-Reproduction and Darwinian Evolution in Autocatalytic Chemical Reaction Systems"

_life, 2021, doi:10.3390/life11040308_

Round 1

Reviewer 1 Report

This is an excellent review. I learned a lot from it. It is very well-written and thorough in describing the literature, from old to new. I make a substantial amount of comments below, because I believe improving it further could make it a reference review on the evolvability of autocatalytic systems in years to come. I look forward to seeing it published. 

A general comment: the authors should consider that many different fields are interested in this topic: Biology, Chemistry, Computational Sciences, Mathematics, Physics, and all the in-betweens. This means that defining specific terms will make this review have a much broader impact. I give some examples below, but advise the authors to take a good agnostic look at the whole text. Following this line of thought, there is one thing missing in this review to broaden its impact: tables. It would seriously enrich the paper to have at least one table that could be readily accessed by readers. The most obvious one I see would summarise section 2, with each type of autocatalysis described there listed, together with e.g. specific characteristics of each, some seminal experimental advances referenced, some ways in which this type of autocatalysis is ahead, and ways it is behind the others. This could come in section "Summary of experimental systems".

- The authors are very thorough with citations, but I miss them earlier in the Introduction. In line 44, the sentence that ends with "the composition of a set of chemical reactions" should have a citation that agrees with this statement, as many people still question if that is indeed "information". There is appropriate bibliography cited afterwards to cite here and support this statement. The same happens in the next line, with "are fairly well established for a template-based system" should have citations right after, which the authors have below and apply here.

- Please define "canonical" Darwinian evolution in the caption to Fig. 1.

- The footnotes strategy is highly questionable. Much is lost and readers will skip these; if any references only appear in the footnotes, that is a problem, I believe. I think MDPI journals in general do not even allow for footnotes. Please check with the editor and consider re-integrating all footnotes in the main text.

- The concept of Hypercycle is introduced very superficially in line 85. The authors should introduce the concept properly here, with the original citation, or not mention it here and just do so later more properly. This will be required for line 185 and onwards.

-lines 336-7. This debate should be expanded upon, not mentioned so quickly. It is somewhat famous within the community that Vasas et al. (ref. 63) started by saying that autocatalytic networks were not evolvable, and stepped back later together with Kauffman to say they were (ref. 12). It is important for the authors to defend their thesis that autocatalytic systems can be units of evolution in broader strokes (note that some biologists still question this), by diving slightly deeper into the general debate between advocates and opponents. This analysis could also read nicely in the future perspectives.

- Lines 422-434 - Reference 91 is the main basis for this whole section, but that gets lost. When reading line 429, it seems that the study is only now being introduced. Please rewrite this section as to highlight the contribution of that study better in a broader manner.

- It would be good to have a short discussion on the likelihood/significance of random ACS's, given physical-chemical constraints/laws, geochemical constraints, and simulations of the origin of metabolism. E.g. reference 26 showed that the distribution of catalysis in small-molecule autocatalytic networks leading to known biomolecules is very non-random, with some catalysts working on many reactions, and others on only a few. This means that there are constraints imposed by chemistry and geology that may help in predicting and selecting autocatalytic networks.

- lines 616-7 "forming different compositions" lacks a citation. This section on macrocycles is slightly poor - it would be good to start by saying what a macrocycle is (this review will reach an audience unfamiliar with the term) and it is not very clear if it may be possible to obtain autocatalysis of macrocycles without peptides. This is, if the experimental advances achieved with peptides are one of several possible routes, or the only possible one (as it reads now one would think the latter, but I suspect that's not the case).

- In lines 711 and 714, the authors give examples twice of systems where network construction has not yet been demonstrated. It is unclear why this repetition. Please clarify this section, one cannot discern between both statements and examples.

- lines 763-765, statement "should (i) spontaneously emerge from an abiotic chemical mixture (non-catalytic building blocks)" reads slightly poor. The abiotic chemical mixture most likely included catalytic molecules, which may or not have been building blocks.

Finally, a couple of papers that I consider relevant to this review, that the authors may consider (or not) to include:

Wilhelm, T. The smallest chemical reaction system with bistability. BMC Syst Biol 3, 90 (2009). https://doi.org/10.1186/1752-0509-3-90
Hordijk, W.; Naylor, J.; Krasnogor, N.; Fellermann, H. Population Dynamics of Autocatalytic Sets in a Compartmentalized Spatial World. Life 2018, 8, 33. 

Minor

- Please revise the grammar in sentence line 336-337

- If possible, please move Fig. 4 down, to after where it is mentioned

- Line 566 - "As discussed earlier" - please indicate where in parenthesis (which section).

line 571, 576 -  "the" is missing before "reaction" twice.

line 664 - "using which the dynamics"

line 697 - "presence of an underlying a well-connected"

Congratulations for your work!

Reviewer 2 Report

The review is, overall, fine.

The only important inconsistency (major correction) that I detected was in the use of the term 'self-reproduction'. In the first part of the manuscript it is put forward as a 'system property' (i.e., the multiplication of an ACS, provided that it finds the adequate stationary state...) whereas in the second part (where real, empirical systems are described) it is applied directly to molecules (as if this part had been written by different authors). My suggestion would be to use the term 'self-replication' for molecular entities and leave self-reproduction for systems or sets of molecules, as it is initially done.

A second (relatively important) weakness of the review is related to the section on 'inorganic chemistries' (i.e., section 2.3). Authors show an obvious lack of knowledge in that area: a straightforward search for an in-depth review on 'chemical gardens' would help them complete that section. 

Reviewer 3 Report

The present paper is a review of theoretical and experimental works on autocatalytic reaction systems (ACSs) prior to the generation of life.  Since it seems quite difficult to construct a self-reproduction system (SRS) abiotically, it is reasonable to consider ACSs prior to the self-reproduction system.  Thus, I think that the review treating ACSs is useful.  The followings are my comments to improve the review.

  1. As the authors told, it is quite difficult to initiate the first self-reproducing system such as the RNA world from a “messy chemistry”.Still most of the ACSs described in Chapter 2 (Experimental Autocatalytic Systems) are based on such sophisticated molecules as DNAs, RNAs and peptides.  It is important to introduce “less sophisticated systems” as (or prior to) ACSs.  We should consider the first catalytic molecules before the first ACS.  I do not think that it was peptides nor RNAs.  I think that a hint was given in Melvin Calvin’s paper where simple iron complex was a catalyst.  Please check review papers appeared in OLEB (Clay and the origin of life, Trace elements and in chemical evolution).
  2. In the study of origins of life, the origin of homochirality of biomolecules (amino acids and sugars) is a quite important issue, since racemic amino acids / nucleic acids never proceeded to ACSs nor SRSs. It was suggested that (i) small enantiomeric excesses (EEs) were formed at first, and then (ii) EEs were enlarged to near 100% (homochirality).Kenso Soai and coworkers showed that quite small EEs could increase by autocatalytic reactions.  Would you include such topics?
  3. Other minor comments:
  • Page 2, Fig. 1: Three dots appeared in my copy. Please check it.
  • Page 3, line 91: The term “a molecule set” looks no good, since a food is also a molecule.
  • Page 3, line 97: Please give a full spelling of “KEGG”.
  • Page 14, line 487: aromatic products such as amino acids, …, hydroxy acids, …, pyruvate: Most of them are not aromatic molecules.
  • Page 23, Reference 3: Origins of life 2nd Edition

Reviewer 4 Report

Ameta et al. nicely reviewed both of theoretical and experimental advances on our understanding of autocatalytic chemical reaction systems, as a possibly key stage in the origin of life. I think the manuscript is nicely written with substantial argument supported by up-to-date references.

There is only one rather minor suggestion. The whole manuscript seems to be focusing on biogeochemistry perspective of origin of life with extensive discussions focusing on organic chemistry. I certainly agree that organic chemistry is the key to understand the origin of life as life itself is organic. However, geological conditions, such as the geological abundances of organics and catalysts (for example, Ru-catalyst in Fig. 3), pH, temperature (and pressure) etc., may provide important or even severe constraints on the possibility of life’s reproduction and evolution including the autocatalytic chemical reaction systems. I suggest that the authors may acknowledge some of these constraints when they discuss the rather ideal case of self-reproduction and evolution of autocatalytic chemical reaction systems.

Overall, I suggest to publish this manuscript (maybe after acknowledging some geological constraints on this whole organic-soup story).

Round 2

Reviewer 2 Report

Regarding my first point, I am afraid it remains an issue if you consider, for instance, Ghadiri's peptides as 'self-reproducing'. And this is not because of my personal conception of 'self-reproduction', but because of the way in which you introduce/use/elaborate the concept in the first part of the article.

Anyway, I told you what I had to tell you about it. Proceed as you wish.

Similarly, the reason why I bring 'chemical gardens' to the fore is not because I like that area of research, or because I think it is very relevant for origins of life. 

The problem comes from the fact that you write a paper about autocatalysis, you include a section on 'experimental autocatalytic systems' (and, specifically a subsection on 'inorganic chemistries')... and then you dispatch it in a couple of paragraphs, with a single reference, as if there was almost no knowledge/evidence on autocatalytic behaviour in the inorganic world. 

I will not fight, you can keep the section as you like. It will simply show your level of ignorance. If you spend some time reviewing seriously the field of "complex inorganic chemistry", you will perhaps find reasons why that type of autocatalytic behaviour is not interesting/useful for your purposes. But ignoring its existence.... 

So I will accept the manuscript in present form, but not because my suggested amendments have been taken care of by the authors, really.